# Elucidating Genotypic Variation in Quinoa via Multidimensional Agronomic, Physiological, and Biochemical Assessments

**DOI:** 10.3390/plants14152332

**Published:** 2025-07-28

**Authors:** Samreen Nazeer, Muhammad Zubair Akram

**Affiliations:** 1Department of Food and Drug, University of Parma, Viale Parco Area delle Scienze 27/A, 43124 Parma, Italy; samreen.nazeer@unipr.it; 2Department of Agricultural, Food and Environmental Sciences (DSA3), University of Perugia, Borgo XX Giugno, 74, 06121 Perugia, Italy

**Keywords:** *Chenopodium quinoa* Willd., germplasm diversification, morpho-physiological attributes, adaptability, climate-proof plant, abiotic stress resilience

## Abstract

Quinoa (*Chenopodium quinoa* Willd.) has emerged as a climate-resilient, nutrient-dense crop with increasing global popularity because of its adaptability under current environmental variations. To address the limited understanding of quinoa’s genotypic performance under local agro-environmental conditions, this study hypothesized that elite genotypes would exhibit significant variation in agronomic, physiological, and biochemical traits. This study aimed to elucidate genotypic variability among 23 elite quinoa lines under field conditions in Faisalabad, Pakistan, using a multidimensional framework that integrated phenological, physiological, biochemical, root developmental, and yield-related attributes. The results revealed that significant variation was observed across all measured parameters, highlighting the diverse adaptive strategies and functional capacities among the tested genotypes. More specifically, genotypes Q4, Q11, Q15, and Q126 demonstrated superior agronomic potential and canopy-level physiological efficiencies, including high biomass accumulation, low infrared canopy temperatures and sustained NDVI values. Moreover, Q9 and Q52 showed enhanced accumulation of antioxidant compounds such as phenolics and anthocyanins, suggesting potential for functional food applications and breeding program for improving these traits in high-yielding varieties. Furthermore, root trait analysis revealed Q15, Q24, and Q82 with well-developed root systems, suggesting efficient resource acquisition and sufficient support for above-ground plant parts. Moreover, principal component analysis further clarified genotype clustering based on trait synergistic effects. These findings support the use of multidimensional phenotyping to identify ideotypes with high yield potential, physiological efficiency and nutritional value. The study provides a foundational basis for quinoa improvement programs targeting climate adaptability and quality enhancement.

## 1. Introduction

The escalating demand for sustainable and nutritionally rich crops has drawn global attention from both researchers and stakeholders toward quinoa (*Chenopodium quinoa* Willd.) due to its nutritional, functional, and climate-resilience properties, positioning it at the forefront of crop diversification strategies [1,2]. Because of these characteristics, the United Nations (UN) declared 2013 the “International Year of Quinoa” recognizing its potential to combat poverty and enhance global food security [1]. Since then, quinoa has evolved from a traditional Andean crop into a globally recognized functional food, celebrated for its exceptional ability to withstand environmental stress conditions [2,3] especially heat [4,5], salinity [6,7], and drought [8,9,10,11]. Given the increasing unpredictability of climate systems, there is a pressing need to identify the crops capable of maintaining growth and productivity under deteriorating climatic conditions. In that context, quinoa stands out as a strategic model for future food security systems because of its physiological plasticity, broad genetic diversity, and remarkable ability to thrive in marginal environments [12].

Recently, efforts have intensified to expand quinoa cultivation beyond the Andean region emphasizing the importance of genotype-environment interaction in determining crop performance. These interactions often manifest as complex physiological and metabolic adjustments that underpin stress responses, resource-use efficiency and yield stability [13]. In this context, genotypic variability plays a crucial role in enhancing both agronomic efficiency and resilience traits under contrasting environmental and agro-ecological regimes [14]. However, despite the growing interest in quinoa improvement, there remains a gap in integrated phenotypic evaluations that effectively link above-ground performance to underlying functional traits across multiple dimensions. Between 1984 and 2014, quinoa cultivation widely diffuses from 8 to 75 countries worldwide, including Canada, UK, Denmark, China, Italy, India, Netherlands, Brazile, Cuba, and Pakistan [15]. Originating from the Andean highlands, quinoa has been cultivated for approximately 7000 years as a key staple food for that region. Bolivia and Peru, situated within its center of origin, continue to represent the primary hubs of global quinoa production and genetic diversity [16]. It is an annual herbaceous dicotyledonous species characterized by a hollow erect stem, goosefoot-shaped leaves, and contains two types of inflorescence: amaranthiform and glomerulate [17]. It has a balanced nutritional profile better than other cereals with superior protein quality, i.e., rich in all essential amino acids, vitamins, minerals, and fiber. Additionally, a handsome concentration of bioactive compounds, particularly ferulic and sinapinic acids, flavanols, kaempferol, and quercetin, are present in the quinoa [18]. Another important feature that helps its global attention is the absence of gluten and a low glycemic index (GI) [19].

State-of-the-art techniques in plant phenotyping emphasize the importance of adopting multidimensional frameworks to evaluate crop performance more systemically. These approaches incorporate morpho-physiological, structural, and biochemical descriptors to analyze the complexity of plant adaptation and resource acquisition strategies [20,21]. For quinoa, such multidimensional assessments are especially valuable given the crop’s capacity to adjust growth, photosynthetic activity, and secondary metabolism in response to fluctuating environmental cues. The variations captured with these approaches aids in identification of stable and high-performing genotypes that maintains its productivity with highly vulnerable climatic aspects without compromising its nutritional profile [22].

Studies conducted under diverse field conditions have demonstrated a significant genotypic difference in quinoa in terms of its yield potential, phenological stability, and stress tolerance indices. However, many of these studies have predominantly focused on single-trait categories or relied on conventional selection indices that fail to include physio-chemical robustness [15,23,24,25]. The integration of high-resolution physiological measurements including Infrared Canopy Temperature (ICT), Normalized Difference Vegetation Index (NDVI), and targeted biochemical analyses has proven to be a potential tool in identifying genotypes with superior performance, as it allows for the early detection of functional traits associated with tolerance mechanisms, metabolic regulation, and source–sink dynamics [26]. Both attributes, including ICT and NDVI, from which ICT provide insights into plant transpiration and temperature regulation [27], while NDVI indicated about vegetation density and photosynthetic activity [28], are vital in helping to distinguish high-yielding and vigorous genotypes.

Moreover, recent advances in multidimensional trait analysis, including multivariate statistical methods and trait-based modeling, have enriched our ability to characterize notable genotype clustering, trait interdependencies, and predictive indicators of performance. These approaches are now considered fundamental in boosting the ideotypes selection under current climate-adaptive agriculture [29]. These methods in quinoa can facilitate the identification of elite genotypes from huge germplasm diversity within quinoa genotypes, with desirable combinations of agronomic features, physiological efficiency and biochemical integrity, an outcome that is not achievable through univariate assessments alone.

Quinoa has been successfully introduced and cultivated in Pakistan since 2009, demonstrating considerable adaptability to the country’s agro-ecological conditions. Following extensive agronomic evaluations and multi-location yield trials, the first locally adapted variety, UAFQ7, which originates from New Mexico, USA, was officially released in 2019 [17]. Despite this progress, quinoa remains a relatively novel crop in the region and there is a continued need to explore the genetic diversity within available germplasm to identify genotypes with superior performance potential. To address this, the present study was designed to broaden the genotypic base by evaluating elite quinoa lines with the objective of identifying promising candidates for future varietal development. The study adopts a comprehensive, multidimensional assessment framework integrating productivity measures, adaptive functional traits, and key biochemical attributes to elucidate genotypic variations under the agro-environmental conditions of Faisalabad. We hypothesize that distinct quinoa genotypes will exhibit significant variability across agronomic, physiological, and biochemical aspects, enabling the identification of superior lines with enhanced adaptability and productivity in local conditions.

## 2. Results

### 2.1. Phenological Development

During the experiment, phenological stages were recorded when more than 50% of the plants had reached the respective stage. Among the quinoa genotypes tested, Q82 and Q24 exhibited the shortest crop cycles, completing their life cycle in 124 days after sowing (DAS). In contrast, genotype Q50 required the longest duration to reach the physiological maturity taking 170 DAS. Notably, Q82 was the earliest to reach the flowering stage among all genotypes. Among 23 quinoa genotypes evaluated, 19 reached the flowering stage, while 15 attained maturities earlier than the standard check variety, i.e., UAFQ7. Detailed results are presented in Table 1.

### 2.2. Biochemical Parameters at Flowering

At flowering (on average = 80 DAS), the leaf samples were collected for the analysis of photosynthetic pigments, including chlorophyll a (*Chl a*), b (*Chl b*), total (*Tol Chl*), carotenoids, total phenolic and anthocyanin contents. The results revealed significant differences (*p* < 0.001) among the tested genotypes for all measured parameters. Genotypes Q4 and Q30 recorded the highest concentrations of *Chl a*, while Q126 exhibited the highest levels of *Chl b* and *Total Chl*. Genotypes Q52 showed the highest carotenoid concentration in its leaves. Additionally, Q9 recorded the greatest values for both phenolics and anthocyanin concentrations. Interestingly, compared to standard check variety UAFQ7, 22 genotypes produced higher values for *Chl a*, 19 showed higher for *Chl b*, 20 had increased *tot Chl*, 6 surpassed in carotenoids, 13 revealed higher values for phenolics and 4 genotypes exhibited higher anthocyanin values. The detailed results are shown in Table 2.

### 2.3. Root Scanning Attributes at Flowering

The results revealed significant variations (*p* < 0.001) among the genotypes tested for the root developmental traits except root volume which was not statistically significant (Table 3). The longest root length (RL) and projected area (PA) were recorded in the standard check variety UAFQ7, although genotype Q15 exhibited comparable values (not-significantly different from UAFQ7) for both parameters. The greatest surface area (SA) was observed in Q15, while Q56 recorded the highest average diameter (Avg. D). Importantly, none of the genotypes outperformed UAFQ7 in terms of RL (Figure 1) and PA (Table 3).

However, four genotypes exceed UAFQ7 in SA, while six genotypes showed higher Avg. D than UAFQ7. In particular, Q4, Q15, Q24, and Q82 demonstrated superior root traits overall, with the exception of RL and PA. The detailed results are presented in Table 3.

### 2.4. Infrared Canopy Temperature (ICT)

Following flowering, data on canopy temperature were collected at three time points (*t*_1_: 94 DAS, *t*_2_: 108 DAS, and *t*_3_: 125 DAS) during the experiment. At each time point, significant differences (*p* < 0.001) were observed among the genotypes (Figure 2). Interestingly, the following different genotypes exhibited the highest canopy temperature at each time point: Q22, Q62, and Q82, indicating genotyping variations among genotypes tested and exhibited higher values *t*_1_, *t*_2_, and *t*_3_, respectively. Moreover, 17 genotypes displayed lower infrared canopy temperature values at *t*_1_ and *t*_2_, suggesting superior adaptability to the prevailing agro-ecological conditions may be due to better transpiration cooling, enhanced efficiency of water use, well-regulated stomatal functioning, and better photosynthetic activities. On the contrary, genotypes Q51 and Q67 consistently exhibited higher canopy temperatures than UAFQ7 across all time points, highlighting less adaptability to current environmental conditions.

### 2.5. Normalized Difference Vegetation Index (NDVI)

Similarly to the ICT, the Normalized Difference Vegetation Index (NDVI) was recorded at the same time points: *t*_1_: 94 DAS, *t*_2_: 108 DAS, and *t*_3_: 125 DAS. At all time points, the results revealed significant variation among the tested genotypes (*p* < 0.001; Figure 3). Genotypes Q4, Q27, and Q81 exhibited the highest NDVI values at *t*_1_, *t*_2_, and *t*_3_, respectively. Interestingly, 7, 1, and 14 genotypes displayed higher NDVI values compared to standard check variety UAFQ7 at *t*_1_, *t*_2_, and *t*_3_, respectively. Among them, Q4 consistently outperformed UAFQ7 across all time points, indicating higher chlorophyll, greater biomass accumulation and overall better crop performance. Conversely, Q24, Q81, and Q82 revealed a marked decline in NDVI values over time, suggesting the onset of some stressful conditions. In contrast, genotypes such as Q9, Q15, Q22, Q30, Q45, Q50, Q52, Q122, and Q124 maintained relatively stable NDVI values throughout the observation period, reflecting consistent growth and vigor. Although UAFQ7 showed relatively higher NDVI values at *t*_1_, a reduction was observed at *t*_2_, followed by a pronounced decline at *t*_3_, suggesting early growth advantages but reduced performance at later developmental stages.

### 2.6. Yield-Related Traits at the End of the Experiment

At the end of the experiment, yield-related traits were assessed, including plant height (PH), plant biomass (PB), main panicle length (MPL), main panicle grain yield (MPGY), grain yield (GY), 1000 grain weight (1000 GW), harvest index (HI), and germination percentage (GP). All parameters showed significant variation among the evaluated genotypes (*p* < 0.001; Table 4). More specifically, Q15 exhibited the tallest PH (Figure 4) and recorded the highest GP (Table 4).

Genotype Q11 outperformed other tested genotypes in terms of PB (Table 4), MPGY (Table 4), and GY (Figure 5). Furthermore, Q51 and Q124 displayed the longest main panicle and highest 1000 GW, respectively.

Notably, Q24 showed the shortest crop cycle and achieved the highest harvest index (Table 4), indicating efficient resource allocation to grain production. Among 23 quinoa genotypes tested, 9, 7, 8, 6, 11, 7, 17, and 19 among 23 genotypes surpassed the standard check variety, i.e., UAFQ7 in PH, PB, MPL, TPGY, GY, 1000 GW, HI, and GP, respectively.

### 2.7. Principle Component Analysis (PCA)

A principal component analysis (PCA) was performed to explore the multivariate structure of the physiological, biochemical, and agronomic traits recorded across 23 quinoa genotypes (Figure 6). The first two principal components, F1 and F2, accounted for a cumulative 41.63% of the total variation, with F1 explaining 27.53% and F2 explaining 14.1%. The resulting biplot provides insight into the relationships among variables and the distribution of genotypes based on trait performance. Traits associated with productivity, such as grain yield (GY), main panicle grain yield (MPGY), plant height (PH), plant biomass (PB), and NDVI1 and NDVI2—contributed strongly to PC1. Genotypes Q4, Q6, Q11, Q15, and Q126 were closely aligned with these traits on the positive axis of F1, indicating superior agronomic performance and yield potential under the tested conditions.

On the other hand, F2 was predominantly influenced by physiological variables including chlorophyll a (*Chl a*), chlorophyll b (*Chl b*), total chlorophyll (*Tot. Chl*), root length (RL), surface area (SA), and pigment accumulation. Genotypes such as Q82 and Q126 showed strong associations with these variables, suggesting enhanced photosynthetic efficiency and vegetative growth. Biochemical quality parameters such as phenolics, anthocyanins, and NDVI3 were negatively associated with F1 and/or F2, forming a distinct cluster away from the yield-related traits. Genotypes Q9, Q22, and Q50 were in this direction, indicating their potential for use in breeding programs focused on nutritional quality and stress-related secondary metabolites. In contrast, genotypes Q67, Q62, and Q56 were distributed on the negative axis of F1 and distanced from most major agronomic traits, indicating comparatively lower performance under the investigated conditions.

Based on multivariate distribution, genotypes Q4, Q6, and Q126 emerged as promising candidate genotypes for high-yield selection, combining favorable productivity traits. Genotype Q82 also demonstrated desirable vegetative and pigment traits, while Q9 and Q22 may serve as valuable sources of bioactive compounds for breeding programs targeting antioxidant-rich quinoa varieties.

## 3. Discussion

Using a comprehensive, multidimensional phenotyping framework, this study presents strong evidence of considerable genotypic variation among quinoa genotypes cultivated under field conditions. The results underscore that the performance of quinoa genotypes depends not only on isolated traits, but on the coordinated expression of phenological, physiological, biochemical, and agronomic attributes [30]. The present study demonstrated significant genotypic variability among 23 elite quinoa genotypes in response to local agro-environmental conditions. Specifically, genotypes Q4, Q11, Q15, and Q126 exhibited superior agronomic performance and favorable canopy-level physiological responses, while Q9 and Q52 were characterized by elevated levels of antioxidant compounds, including phenolics and anthocyanins. These results underscore the potential of selected genotypes for use in breeding programs aimed at improving yield, stress adaptability and nutritional quality. This integrated approach proved it was necessary to distinguish high performance genotypes with a balanced trait profile, enabling the selection of leading candidate genotypes for targeted reproductive purposes in alignment with the study’s objectives and underlying hypothesis.

A huge diversification was evident in the genotypes used in the experiment specifically in terms of phenological growth stages. The observed diversity in phonological development, ranging from early- to late-maturing genotypes, reflects a broad adaptive window among the tested genotypes in the present study. However, this diversification alone was not predictive of yield performance, highlighting the importance of interpreting simultaneously the physiological and morphological efficiencies [31]. For example, as demonstrated by Q24, which appeared to be the shortest in the crop cycle, exhibited the highest harvest index. In contrast, genotype Q11, a late-maturing genotype, produced the highest grain yield and above-ground biomass accumulation. The results align with the findings from previous studies, indicating that quinoa productivity is not exclusively dependent on crop cycle duration, but rather is a combination of integrated physiological performance and sour–sink dynamics [12,15,32,33].

Furthermore, genotypes Q4, Q11, Q15, and Q126 demonstrated exceptional agronomic performance, accompanied by the favorable physiological and canopy traits, including lower infrared canopy temperatures and sustained Normalized Difference Vegetation Index (NDVI) values throughout the reproductive growing cycle. These indicators demonstrate efficient energy dissipation, well-regulated transpiration, and stable chlorophyll content, all of which work together to promote greater vegetative vigor and increased assimilate accumulation [34]. These physiological efficiencies align with the performance-enhancing mechanisms reported in the previous open-field phenotyping studies [26,27,28]. Consistent with physiological performance, biochemical evaluations revealed that genotypes exhibited enhanced chlorophyll pigmentation, carotenoids, and phenolic compounds, which corresponded to improved vegetative and, in some cases, reproductive consequences. For example, paired genotypes Q30 and Q126 increased pigment concentrations with vigorous canopy development. On the other hand, genotypes Q9 and Q52 were not the top yielding genotypes, but accumulated significant antioxidant metabolites levels, highlighting their potential for functional food applications [35,36]. Similar results have been revealed in previous studies that identified genotype-specific metabolic signatures contributing to the quinoa nutritional enhancement [15].

Root trait analysis further expanded the differentiation pool among genotypes, particularly highlighting Q15, Q24, and Q82, which exhibited well-developed root systems and elongation even better than standard check variety UAFQ7. Their enhanced root surface area and average diameter facilitate an improvement in resource acquisition and aids in better above-ground biomass development [9,17], also reported for other crops [37]. Although measured under non-limiting soil conditions, the expression of such traits may reflect intrinsic functional strategies that, if consistent across environments, could be exploited for enhanced nutrient-use efficiency and performance stability [13,38]. Multivariate analysis, i.e., Principal Component Analysis (PCA) confirmed the integration of these trait domains, revealing genotype clusters characterized by the simultaneous expression of yield attributes, physiological efficiency, and biochemical traits. Genotypes occupying favorable positions in PCA space, such as Q4 and Q126, highlight the potential to identify ideotypes that combine agronomic performance with physiological and nutritional robustness. Conversely, genotypes exhibiting strength in a single domain (e.g., antioxidant content, i.e., Q9) may be strategically utilized in breeding programs focused on specific value-added traits [8,39,40,41].

The results of this study underscore the need for quinoa breeding programs to adopt a comprehensive strategy for selection by integrating a wide range of morpho-physiological, yield-related, and biochemical traits in addition to traditional yield parameters. The identification of genotypes with consistently favorable performance across multiple trait dimensions underscores the value of integrative phenotyping in capturing complex genotype × trait interactions that drive field-level outcomes [42]. Future research directions should focus on multi-environment field trials to assess the stability of identified trait combinations and genotype rankings under diverse agro-climatic conditions. Evaluating these genotypes across different biotic and abiotic stress gradients, particularly under water-limited, salinity, heavy metal, and high-temperature environments, will provide critical insights into their adaptive plasticity and functional resilience. Furthermore, integrating high-throughput phenotyping platforms with genomic and transcriptomic tools could enable the dissection of genetic loci underlying key traits such as canopy thermal regulation, pigment biosynthesis, and root system architecture. Such genomic insights would facilitate the development of marker-assisted selection pipelines tailored to complex trait integration. Additionally, studies exploring trait heritability and genotype × environment interaction will be essential for developing stable, broadly adaptable quinoa cultivars suited to emerging climate challenges, and nutritional markets [43]. This experiment revealed that quinoa’s performance is driven by closely interconnected trait combinations. Genotypes like Q4, Q11, Q15, and Q126 emerged as strong candidates for varietal improvement aimed at achieving high productivity and enhanced physiological efficiency. Moreover, Q9 and Q52 offer valuable biochemical attributes for quality-focused breeding programs. These findings advocate for a shift toward multidimensional, trait-informed selection strategies in quinoa improvement, ultimately supporting its global expansion as a climate-adaptable and nutritionally rich crop.

## 4. Materials and Methods

### 4.1. Experimental Layout

A field experiment was conducted using 23 elite quinoa genotypes, which were previously selected based on their morphological traits and yield adaptability from an initial pool of 128 lines imported from the USDA [15], during fall–winter 2017–18 at directorate farms, University of Agriculture, Faisalabad, Pakistan (31°7′ N, 73°7′ E, 184.4 m a.s.l., Pakistan). The seeds of quinoa genotypes were collected from the Alternate Crops Lab, Department of Agronomy, University of Agriculture, Faisalabad, Pakistan. Among 23, one variety, i.e., UAFQ7, was already approved as the first quinoa variety in Pakistan, used as standard check variety. Details of genotypes used in the experiment can be found in Table 5.

The soil used in the experiment was sandy loam in texture. Detailed soil properties were mentioned in Table 6. However, the soil is classified as Lyallpur soil series in Haplic Yermosols according to FAO classification.

Ridges were prepared in the field to establish the seedbed, maintaining a row spacing of 75 cm between them. Seeds were manually sown on 23 November 2017 using hand-sowing, with a plant-to-plant spacing of 15 cm distance, following a randomized complete block design (RCBD). Nitrogen, phosphorus, and potassium were applied using urea, diammonium phosphate, and sulfate of potash at 75:50:50 kg/ha, respectively. However, a half nitrogen dose and full phosphate and potassium doses were applied at the time of sowing. The remaining half nitrogen dose was applied at the time of first irrigation. Two hoeing operations were carried out at 45 and 70 days after sowing to manage the weeds and maintain soil aeration. The weather data recorded during the experiment are presented in Table 7.

### 4.2. Plant Measurments

#### 4.2.1. Plant Phenology

Phenological data were recorded by regularly visiting the experimental area when more than 50% of the plants in a plot reached a specific developmental stage, following the scale described by Sosa-Zuniga et al. [44]. The stages were categorized as from S1 to S11 as follows: S1: Emergence, S2: True Leaf, S3: Four leaf, S4: Multiple Leaf, S5: Bud Visible, S6: Bud distinct, S7: Pyramid formation, S8: Flowering, S9: Milking, S10: Seed Set, S11: Maturity.

#### 4.2.2. Biochemical Attributes

Fully expanded mature leaves were collected at S8 and immediately preserved in liquid nitrogen, followed by storage in a biomedical freezer at −80 °C until analysis. These samples were used to quantify the chlorophyll a (*Chl a*: mg/g FW), chlorophyll b (*Chl b*: mg/g FW), total chlorophyll (*Total Chl*: mg/g FW), carotenoids (mg/g FW), total phenolics (µg/g FW), and anthocyanin concentrations in the leaves.

The protocol developed by Nagata and Yamashta [45] was followed for an estimation of *Chl a* and *Chl b* concentrations in the leaves. For that purpose, 0.5 g leaf sample was ground in 10 mL of 80% acetone using pestle and mortar. The ground material was then put in falcon tube and centrifuged for 15 min at 4000 rpm. The resulting supernatant was subjected to take absorbance at 663 for *Chl a*, 645 for *Chl b*, and 470 nm for carotenoids using spectrophotometer (Shimadzu UV 4000, Shimadzu Corporation, Kyoto, Japan).

The total phenolic contents was determined according to the method described by Waterhouse [46]. For that purpose, a 0.5 g leaf sample was ground in 10 mL of 80% acetone using a mortar and pestle. The homogenate was transferred to a Falcon tube and centrifuged at 4000 rpm for 15 min. A 20 µL aliquot of the resulting supernatant was transferred to a test tube, followed by the addition of 1.58 mL distilled water and 100 µL of *Folin*–*Ciocalteu reagent*. After vertexing for 30 s, 300 µL of 20% sodium carbonate (Na_2_CO_3_) was added. The test tubes containing the assay solution were incubated in a water bath at 40 °C for 30 min. Absorbance of both standards and samples was measured at 760 nm using a spectrophotometer (Shimadzu UV4000, Shimadzu Corporation, Kyoto, Japan). Absorbance values were plotted against the corresponding concentrations of gallic acid to derive the regression equation for phenolic determination.

Additionally, the anthocyanin content (mg CGE 100 g^−1^ DW) was measured by using the method developed by Lees and Francis [47]. The absorbance for anthocyanin was set at 532 nm in spectrophotometer (Shimadzu UV4000, Shimadzu Corporation, Kyoto, Japan) and expressed as mg CGE 100 g^−1^ DW (cyanidin-3-glucoside equivalents per 100 g dry weight).

#### 4.2.3. Root Related Traits

Simultaneously with the biochemical analysis, root samples were collected to evaluate root-related traits. For root analysis, careful excavation was carried out to minimize root damage and samples were gently washed under a water stream to remove the adhering soil and debris. Cleaned roots were scanned using a root scanner operated with Epson scan 2 software. The resulting images were analyzed using WinRhizo companion software v7.6.5 to determine root morphological parameters, including root length (RL: cm), projected area PA: (cm^2^), surface area (SA: cm^2^), volume (Vol: cm^3^), and average diameter (Avg. D: mm).

#### 4.2.4. Physiological Attributes

Infrared Canopy Temperature (ICT: °C) and Normalized Difference Vegetation Index (NDVI) were recorded at three different time points following flowering: *t*_1_: 94 DAS, *t*_2_: 108 DAS, and *t*_3_: 125 DAS. For that purpose, an infrared thermometer (RAY, STEP Systems GmbH, Nuremberg, Germany) was used to record the ICT, while NDVI was measured with a Handheld GreenSeeker Optical Sensor Unit (Model-505). All measurements were conducted between 9:00 AM and 11:00 AM to ensure consistency and minimize diurnal variation.

#### 4.2.5. Plant Growth and Yield-Related Attributes

At the end of the experiment, six plants were randomly selected from each plot to assess growth and yield-related attributes. The parameters measured included plant height (PH: cm), plant biomass (PB: g), total grain yield (GY: g), main panicle length (MPL: cm), and main panicle grain yield (MPGY: g), thousand grain weight (1000 GW: g), harvest index (HI), and germination percentage (GP: %). The PH and MPL were measured by using a meter rod. The PB was determined by drying the harvested above-ground biomass in an oven at 70 °C until the constant weight was achieved and weighed at weight balance. The GY was observed by taking the weight of all the seeds obtained from the terminal panicle (main panicle) as well as from sub-panicles, however MPGY only included the weight of seeds from the terminal panicle (main panicle). For determining 1000 GW, seeds were counted using a seed counter (Model #14100011, Pfeuffer GmbH, Kitzingen, Germany) and then weighed. Finally, the HI was calculated as the ratio of economic (GY) to biological yield (PB).

### 4.3. Statistical Analysis

All experimental data were tested for normality and homogeneity of variances prior to statistical analysis. One-way analysis of variance (ANOVA) was conducted under a randomized complete block design (RCBD) to evaluate the effects of different tested genotypes used in the experiment. When significant differences among means were detected, Tukey’s Honest Significant Difference (HSD) test was applied for multiple comparisons at a significance level of *p* < 0.05%. Additionally, principal component analysis (PCA) was performed to explore the relationships among the measured traits and to visualize varietal responses and their variations. All statistical analyses including PCA were carried out using RStudio software v4.2.0 [48].

## 5. Conclusions

This present investigation highlighted significant genotypic variability among quinoa genotypes by employing an integrated phenotyping approach that combined agronomic, physiological, biochemical, and root traits under open-field conditions. Genotypes such as Q4, Q11, Q15, and Q126 demonstrated superior growth performance across multiple dimensions, indicating their potential for high-yield, climate-resilient cultivation. On the other hand, Q9 and Q52 stood out for their rich antioxidant profiles, suggesting value in nutritional breeding programs. The standard check variety UAFQ7 was outperformed by many elite quinoa genotypes in key traits, including yield, pigmentation, i.e., chlorophyll content, physiology, i.e., NDVI and root system development, highlighting the superior adaptability and functional efficiency of selected lines such as Q4, Q11, Q15, and Q126. This comparison underscores the potential of the evaluated germplasm to enrich or complement existing variety, i.e., UAFQ7 under similar agro-ecological conditions. The use of Infrared Canopy Temperature (ICT) and Normalized Difference Vegetation Index (NDVI) effectively differentiated genotypes with superior stress adaptability and photosynthetic efficiency. Moreover, root traits further highlighted below-ground strategies that support resource-use efficiency. These findings emphasize the need for breeding programs to adopt multidimensional selection frameworks rather than relying solely on yield metrics. Future studies should focus on multi-environment trials, explore trait heritability and integrate genomic tools to uncover the genetic basis of complex traits, enabling the development of robust quinoa ideotypes tailored for both productivity and nutritional quality in diverse agro-climatic regions.

## Figures and Tables

**Figure 1 plants-14-02332-f001:**
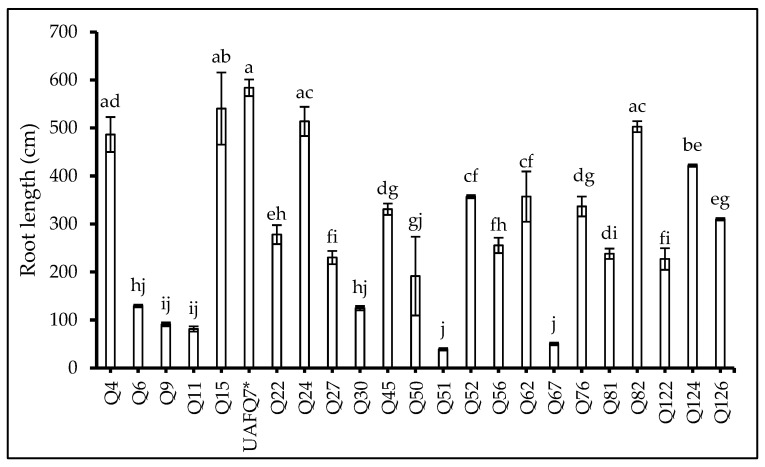
Root length (RL: cm) of 23 elite quinoa genotypes (Q4-Q126) recorded at flowering. Values are means (*n* = 3) ± S.E. In each graph, different letters above the bars indicate significant differences among treatments (*p* < 0.05, Tukey’s test). F test significant at *p* < 0.001. Standard check variety—UAFQ7*.

**Figure 2 plants-14-02332-f002:**
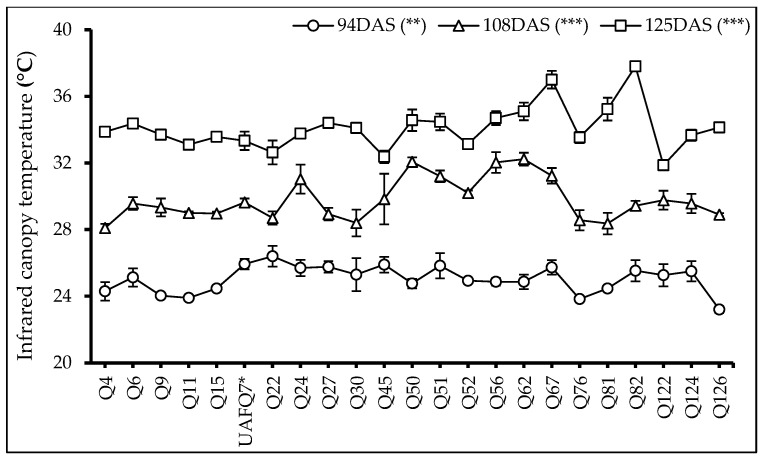
Infrared Canopy Temperature (ICT: °C) of 23 elite quinoa genotypes (Q4-Q126) recorded at three time points, i.e., *t*_1_: 94 DAS, *t*_2_: 108 DAS, and *t*_3_: 125 DAS. Values are means (*n* = 3) ± S.E. Stars after each data label indicate significant differences among treatments (*p* < 0.05, Tukey’s test). F test significant at **—*p* < 0.01, ***—*p* < 0.001. Standard check variety—UAFQ7*.

**Figure 3 plants-14-02332-f003:**
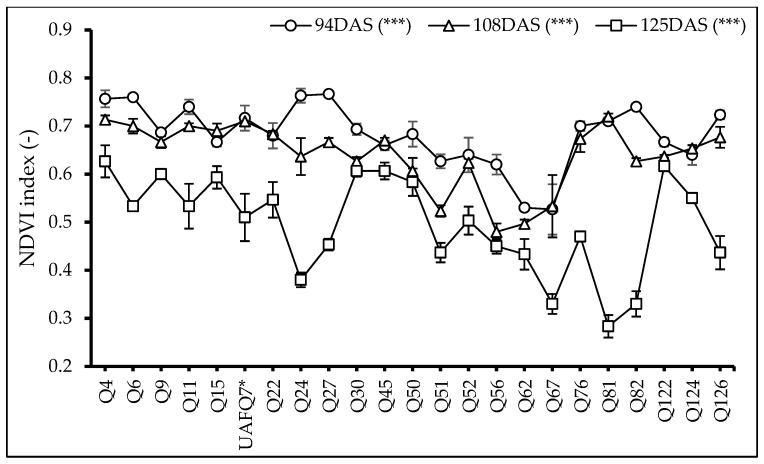
Normalized Difference Vegetation Index (NDVI) of 23 elite quinoa genotypes (Q4-Q126) recorded at three time points i.e., *t*_1_: 94 DAS, *t*_2_: 108 DAS, and *t*_3_: 125 DAS. Values are means (*n* = 3) ± S.E. Stars after each data label indicate significant differences among treatments (*p* < 0.05, Tukey’s test). F test significant at ***—*p* < 0.001. Standard check variety—UAFQ7*.

**Figure 4 plants-14-02332-f004:**
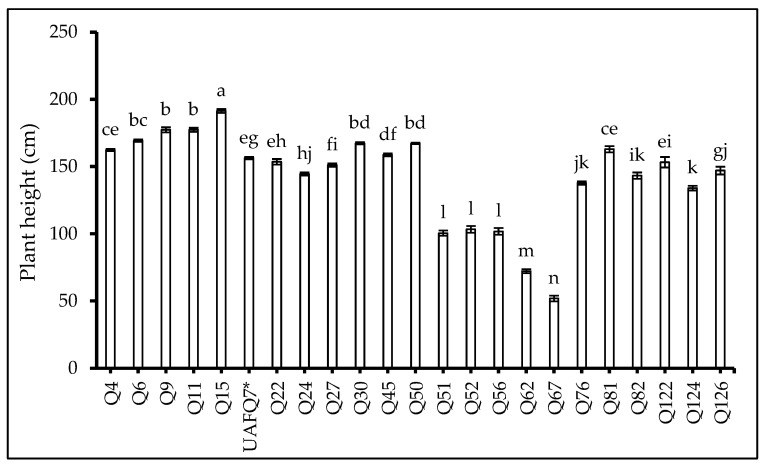
Plant height (PH: cm) of 23 elite quinoa genotypes (Q4–Q126) recorded at flowering. Values are means (*n* = 3) ± S.E. In each graph, different letters above the bars indicate significant differences among treatments (*p* < 0.05, Tukey’s test). F test significant at *p* < 0.001. Standard check variety—UAFQ7*.

**Figure 5 plants-14-02332-f005:**
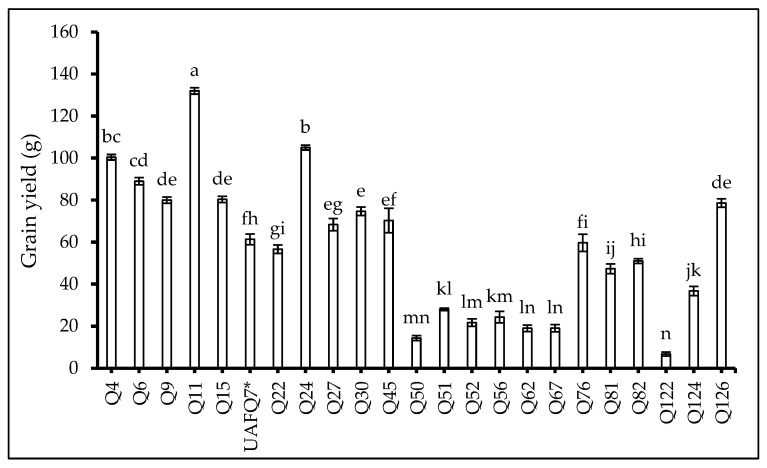
Grain yield (GY: cm) of 23 elite quinoa genotypes (Q4-Q126) recorded at flowering. Values are means (*n* = 3) ± S.E. In each graph, different letters above the bars indicate significant differences among treatments (*p* < 0.05, Tukey’s test). F test significant at *p* < 0.001. Standard check variety—UAFQ7*.

**Figure 6 plants-14-02332-f006:**
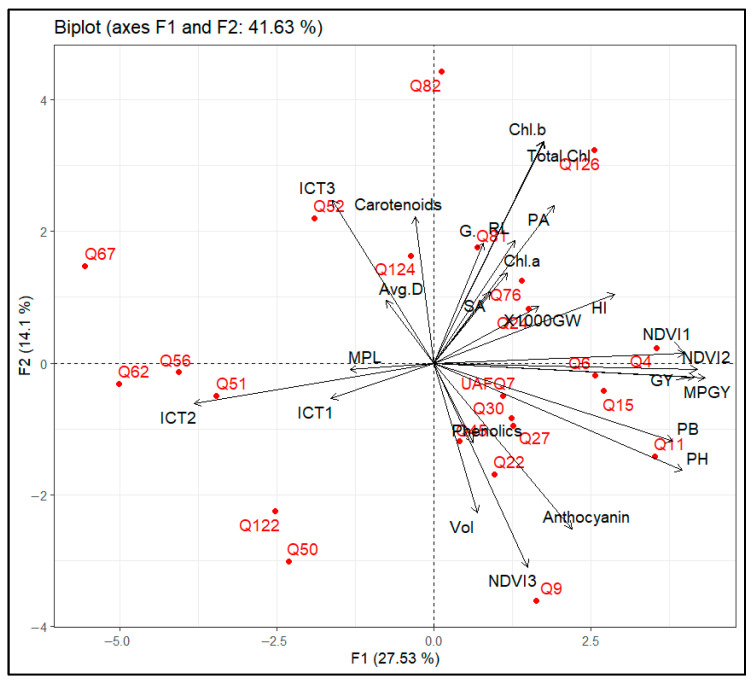
Correlation biplot derived from Principal Component Analysis (PCA) of 23 quinoa genotypes based on agronomic, physiological, and biochemical traits. Chlorophyll a—*Chl a*; chlorophyll b—*Chl b*; total chlorophyll—*Total Chl*; Root length—RL; root surface area—SA, root projected area—PA, root volume—Vol; root average diameter—Avg. D; Infrared Canopy Temperature—ICT, at 94 (ICT1), 108 (ICT2), and 125 (ICT3) DAS; Normalized Difference Vegetation Index—NDVI, at 94 (NDVI1), 108 (NDVI2) and 125 (NDVI3) DAS; plant height— PH; plant biomass—PB, main panicle length—MPL; main panicle grain yield—MPGY, thousand grain weight—1000 GW, harvest index—HI. Red dots: quinoa genotypes Q4–Q126. Standard check variety—UAFQ7*.

**Table 1 plants-14-02332-t001:** Phenological distribution of 23 quinoa genotypes.

Genotypes	S1	S2	S3	S4	S5	S6	S7	S8	S9	S10	S11
Q4	5	6	14	24	44	57	70	81	110	125	138
Q6	6	8	13	21	46	53	64	78	106	124	146
Q9	6	7	14	24	46	55	66	80	115	132	143
Q11	7	6	13	24	47	55	68	80	105	124	147
Q15	5	6	14	23	44	53	65	79	113	128	145
UAFQ7*	5	6	14	23	51	58	68	78	110	126	145
Q22	5	7	15	23	48	59	70	82	108	122	145
Q24	5	6	14	24	48	58	70	82	102	115	128
Q27	6	7	14	21	45	56	68	79	104	126	146
Q30	6	7	15	25	46	59	68	79	115	129	144
Q45	5	6	14	24	57	65	76	87	110	125	145
Q50	6	6	13	24	55	61	68	78	107	142	170
Q51	7	7	15	24	46	58	68	79	111	128	145
Q52	6	7	13	23	44	58	69	82	106	125	143
Q56	6	4	14	24	45	57	67	77	108	122	142
Q62	6	7	14	22	44	57	70	79	112	130	145
Q67	6	4	14	24	45	55	68	81	108	128	146
Q76	6	7	14	24	55	68	79	89	115	130	147
Q81	6	6	14	25	57	66	77	87	105	127	144
Q82	6	7	14	24	48	64	73	85	106	129	128
Q122	7	6	15	25	51	62	75	86	104	135	149
Q124	6	7	15	22	54	57	69	81	117	132	145
Q126	6	7	15	22	55	65	77	84	115	135	146

S1—Emergence, S2—True Leaf, S3—Four leaf, S4—Multiple Leaf, S5—Bud Visible, S6—Bud distinct, S7—Pyramid, S8—Flowering, S9—Milking, S10—Seed Set, S11—Days to Maturity. Standard check variety—UAFQ7*.

**Table 2 plants-14-02332-t002:** Chlorophyll a (*Chl a*: mg/g FW), b (*Chl b*: mg/g FW), total chlorophyll (*Total Chl*: mg/g FW), carotenoids (mg/g FW), phenolics (mg/g FW), and anthocyanin (mg CGE 100 g^−1^ DW) contents of 23 quinoa genotypes recorded at flowering.

Genotypes	*Chl a* (mg/g FW)	*Chl b* (mg/g FW)	*Total Chl* (mg/g FW)	Carotenoids (mg/g FW)	Phenolics (µg/g FW)	Anthocyanin (mg CGE 100 g^−1^ DW)
Q4	0.045 ab	0.243 ac	0.288 ac	3.354 ab	0.701 ac	0.584 gi
Q6	0.034 cg	0.447 ac	0.482 ac	2.884 ab	0.586 bc	1.144 ac
Q9	0.031 dh	0.150 bc	0.181 bc	2.727 b	0.825 a	1.287 a
Q11	0.030 eh	0.266 ac	0.296 ac	3.336 ab	0.655 ac	1.130 ad
Q15	0.036 af	0.218 ac	0.255 ac	3.701 ab	0.673 ac	1.215 a
UAFQ7*	0.026 gh	0.165 bc	0.191 bc	3.688 ab	0.684 ac	1.149 ac
Q22	0.036 af	0.221 ac	0.257 ac	3.092 ab	0.659 ac	1.212 a
Q24	0.027 gh	0.202 ac	0.228 ac	3.526 ab	0.646 ac	1.180 ab
Q27	0.032 dg	0.219 ac	0.251 ac	3.153 ab	0.809 ab	1.085 ae
Q30	0.045 a	0.321 ac	0.366 ac	3.330 ab	0.683 ac	1.063 af
Q45	0.039 ae	0.276 ac	0.315 ac	2.541 b	0.753 ac	0.799 di
Q50	0.034 cg	0.159 bc	0.193 bc	2.991 ab	0.799 ac	1.204 a
Q51	0.034 cg	0.354 ac	0.388 ac	2.831 b	0.712 ac	0.788 ei
Q52	0.043 ac	0.379 ac	0.422 ac	4.915 a	0.748 ac	0.517 hi
Q56	0.034 cg	0.051 c	0.085 c	3.388 ab	0.564 c	0.671 gi
Q62	0.022 h	0.056 c	0.078 c	3.267 ab	0.711 ac	0.807 di
Q67	0.034 cg	0.219 ac	0.253 ac	3.955 ab	0.599 ac	0.699 gi
Q76	0.037 af	0.564 ab	0.600 ab	2.771 b	0.694 ac	0.745 fi
Q81	0.038 ae	0.388 ac	0.425 ac	3.937 ab	0.717 ac	0.841 ch
Q82	0.039 ad	0.576 ab	0.615 ab	2.648 b	0.713 ac	0.500 i
Q122	0.028 fh	0.081 c	0.109 c	3.183 ab	0.644 ac	0.741 fi
Q124	0.036 bf	0.525 ac	0.561 ac	3.873 ab	0.657 ac	0.854 bh
Q126	0.033 dg	0.669 a	0.702 a	4.508 ab	0.687 ac	1.124 ae
*Level of significance*						
	***	***	***	**	**	***

The different letters followed by values in the same column indicate differences among treatments according to Tukey’s HSD. Values are mean (*n* = 3), F-test significant at **: (*p* < 0.01), ***: (*p* < 0.001). Standard check variety—UAFQ7*.

**Table 3 plants-14-02332-t003:** Surface area (SA: cm^2^), projected area (PA: cm^2^), volume (Vol: cm^3^) and average diameter (Avg. D: mm) of 23 quinoa genotypes recorded at flowering.

Genotypes	SA (cm^2^)	PA (cm^2^)	Vol (cm^3^)	Avg. D (mm)
Q4	94.2 a	32.1 ab	1.453	0.550 d
Q6	36.3 gh	25.1 ag	1.717	0.823 cd
Q9	53.5 ef	14.1 eg	1.691	0.934 bd
Q11	34.2 gh	19.1 bg	1.670	1.782 a
Q15	95.5 a	33.3 a	1.940	1.837 a
UAFQ7*	84.5 ac	33.9 a	1.547	1.837 a
Q22	59.8 df	23.9 ag	1.790	1.620 ab
Q24	93.8 a	32.7 a	1.530	1.647 a
Q27	61.3 de	23.5 ag	1.610	0.857 cd
Q30	44.7 fg	17.0 cg	1.813	1.670 a
Q45	27.0 h	26.5 af	1.640	1.773 a
Q50	54.4 ef	12.7 g	1.833	1.723 a
Q51	33.2 gh	12.1 g	1.533	0.071 a
Q52	60.0 df	31.3 ab	1.443	1.573 ab
Q56	68.6 ce	21.8 ag	1.797	1.947 a
Q62	75.7 bd	24.7 ag	1.560	1.807 a
Q67	24.7 h	16.3 dg	1.360	1.817 a
Q76	66.6 de	29.6 ad	2.163	1.517 ac
Q81	44.2 fg	27.5 ae	1.773	1.757 a
Q82	95.2 a	30.1 ac	1.067	1.753 a
Q122	33.8 gh	13.1 fg	1.527	1.747 a
Q124	85.8 ab	27.9 ad	1.773	1.893 a
Q126	28.6 gh	20.8 ag	1.187	1.717 a
*Level of significance*				
	***	***	ns	***

The different letters followed by values in the same column indicate differences among treatments according to Tukey’s HSD. Values are mean (*n* = 3), F-test significant at ***: (*p* < 0.001), ns: not significant. Standard check variety—UAFQ7*.

**Table 4 plants-14-02332-t004:** Plant biomass (PB: g), main panicle length (MPL: cm), and yield (MPGY: g), thousand grain weight (1000 GW: g), harvest index (HI), and germination percentage (GP: %) of 23 quinoa genotypes recorded at the end of the experiment.

Genotypes	PB (g)	MPL (cm)	MPGY (g)	1000 GW (g)	HI (-)	GP (%)
Q4	174.7 bd	31.7 bd	71.8 ab	0.567 ab	0.576 ac	72.0 j
Q6	172.9 be	34.0 bd	67.9 b	0.467 ab	0.515 ad	98.0 ab
Q9	195.9 ab	26.0 d	54.5 c	0.400 b	0.408 cf	84.0 hi
Q11	216.3 a	34.0 bd	80.8 a	0.533 ab	0.610 ab	94.0 cd
Q15	193.8 ac	29.0 cd	52.9 cd	0.467 ab	0.415 cf	100.0 a
UAFQ7*	158.3 df	35.3 bd	49.6 ce	0.467 ab	0.388 dg	86.0 gh
Q22	177.4 bd	31.3 bd	46.5 cg	0.467 ab	0.319 eg	90.0 ef
Q24	156.4 dg	26.3 d	39.9 ei	0.500 ab	0.672 a	100.0 a
Q27	135.7 ei	27.0 d	43.7 dh	0.500 ab	0.504 ad	88.0 fg
Q30	168.4 df	34.7 bd	39.2 fi	0.500 ab	0.444 be	96.0 bc
Q45	153.7 dh	35.3 bd	31.9 ik	0.367 b	0.458 be	96.0 bc
Q50	141.1 dh	39.7 ab	7.2 m	0.433 ab	0.102 h	84.0 hi
Q51	98.9 ij	46.3 a	23.8 kl	0.400 b	0.283 eg	98.0 ab
Q52	98.9 ij	38.3 ac	13.8 lm	0.467 ab	0.220 gh	92.0 de
Q56	100.4 ij	35.3 bd	11.3 m	0.533 ab	0.242 fh	94.0 cd
Q62	73.4 jk	32.0 bd	5.4 m	0.500 ab	0.260 fh	88.0 fg
Q67	36.7 kl	28.0 d	7.7 m	0.333 b	0.523 ad	82.0 i
Q76	104.7 jk	35.7 bd	39.9 ei	0.400 b	0.571 ac	92.0 de
Q81	130.6 fi	28.0 d	34.8 hj	0.367 b	0.363 dg	98.0 ab
Q82	140.1 dh	31.7 bd	37.4 gi	0.433 ab	0.364 dg	100.0 a
Q122	19.4 l	28.3 bd	4.6 m	0.467 ab	0.383 dg	82.0 i
Q124	116.1 hi	38.3 ac	26.2 jk	0.467 ab	0.316 eg	94.0 cd
Q126	120.1 gi	29.0 cd	47.9 cf	0.733 a	0.655 a	96.0 bc
*Level of significance*						
	***	***	***	*	***	***

The different letters followed by values in the same column indicate differences among treatments according to Tukey’s HSD. Values are mean (*n* = 3), F-test significant at *: (*p* < 0.05), ***: (*p* < 0.001). Standard check variety—UAFQ7*.

**Table 5 plants-14-02332-t005:** Details of elite quinoa genotypes used in the experiment previously imported from USDA.

Genotypes Local Coding	Origin
Q4	United States, New Mexico
Q6	United States, New Mexico
Q9	United States, New Mexico
Q11	United States, New Mexico
Q15	United States, New Mexico
UAFQ7*	United States, New Mexico (Approved quinoa variety in Pakistan)
Q22	United States, New Mexico
Q24	United States, New Mexico
Q27	Bolivia
Q30	United States, New Mexico
Q45	Chile
Q50	Bolivia, La Paz
Q51	Bolivia, La Paz
Q52	Bolivia, La Paz
Q56	Peru
Q62	Peru
Q67	Peru
Q76	Chile, Los Lagos
Q81	Chile, Los Lagos
Q82	Bolivia, Oruro
Q122	Bolivia, La Paz
Q124	Bolivia, La Paz
Q126	United States, Colorado

* Standard check variety.

**Table 6 plants-14-02332-t006:** Soil properties used in the experiment.

Soil Properties	Value
pH	6.8 to 7.3
Electric conductivity	3.62–3.72 dS/m
Organic matter	0.88–0.91%
Total Nitrogen	0.07–0.08%
Exchangeable Potassium	175–177 ppm
Available Phosphorous	4.44–4.60 ppm

**Table 7 plants-14-02332-t007:** Weather data during the course of the experiment.

Months	Temperature			Relative Humidity	Total Rainfall	Sunshine Radiation	Wind Speed
	Max.	Min.	Mean				
	°C	°C	°C	%	mm	h	km/h
November	24.1	11.8	18	84.6	1.5	3.7	1.9
December	22	6.7	14.4	69.3	4.2	6	2.4
January	21.5	5.5	13.5	75.9	0	6.4	3.5
February	24	9.5	16.7	73.3	9.5	6.5	3.8
March	32.2	16.4	23.8	61.4	12.5	8.6	5.2
April	36.8	20.8	28.8	47.3	7.9	9.1	3.1
May	40.3	23.7	32	29.8	21.6	8.6	3.4

Data were collected from Agriculture Metrological Cell, Department of Agronomy, University of Agriculture, Faisalabad, Pakistan.

## Data Availability

The data presented in this study are available upon request from the corresponding authors.

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
