# Peer review of "Elucidating Genotypic Variation in Quinoa via Multidimensional Agronomic, Physiological, and Biochemical Assessments"

_plants, 2025, doi:10.3390/plants14152332_

Round 1

Reviewer 1 Report

Comments and Suggestions for Authors

Quinoa (Chenopodium quinoa Willd.) is an important food crop, whose requirement increase year by year due to its climate-resilience and nutrient-rich. Therefore, elucidating its genotypic variation and selecting high-yield variety are importance for quinoa industry. This manuscript elucidated that genotypic variation in quinoa by agronomic, physiological, and biochemical assessments, it is important to obtain the high-yield variety with climate-resilience and nutrient-rich. However, I think the following comments should be considered.

(1) ABSTRACT: I think the research GAP or hypothesis should be clearly and accurately stated in the ABSTRACT.

(2) INTRODUCTION: OK.

(3) METHODS: In biochemical parameter assessment, the leaf age of leaf samples, such as the young, maturation, or old, should be clearly stated. Also, in general, the samples should be stored in a biomedical freezer at -80°C, NOT -30°C, please check it! In addition, line 367, the unit of anthocyanin in the leaves should be appeared. Besides, the subtitle "4.2.4" might be " Plant growth and yield-related attributes ", NOT " Planth growth and yield-related attributes ", please check it!.

(4) RESULTS: First, the subtitles "2.2 Biochemical analysis ", "2.3. Root scanning ", and "2.3. Root scanning " are all METHODS, NOT RESULTS, please revise! Also, tables 2, 4, 5, and 6, there is not the level of significance "ns: not significant", which should be deleted. In addition, in tables 2, 3, 4, 5, and 6, the level of significance was indicated by the different letters in the same column, so I think that the level of significance "***, **" did not presented. Last, in leaf samples, the content of chlorophyll a (Chl a: mg/g FW), b (Chl b: mg/g FW), and total chlorophyll (Total Chl: mg/g FW) was so lower, and even lower than that of carotenoids (mg/g FW) and phenolics (mg/g FW), please check them! Similarly, in table 2, the unit of anthocyanin in the leaves should be added.

(5) DISCUSSION: In the first paragraph, the main findings should be clearly summarized, and then separately discussed.

(6) LANGUAGE: LANGUAGE should be carefully checked and improved.

Comments on the Quality of English Language

above

Author Response

Quinoa (Chenopodium quinoa Willd.) is an important food crop, whose requirement increase year by year due to its climate-resilience and nutrient-rich. Therefore, elucidating its genotypic variation and selecting high-yield variety are importance for quinoa industry. This manuscript elucidated that genotypic variation in quinoa by agronomic, physiological, and biochemical assessments, it is important to obtain the high-yield variety with climate-resilience and nutrient-rich. However, I think the following comments should be considered.

Response: We thank the reviewer for their constructive and encouraging comments regarding the importance of quinoa as a climate-resilient and nutrient-rich crop, and the relevance of our study. We agree that understanding genotypic variation is essential for selecting high-yielding and resilient varieties. In this study, we aimed to provide a comprehensive evaluation of quinoa genotypes using agronomic, physiological, and biochemical parameters to support breeding and cultivation strategies under changing climatic conditions.

We appreciate the specific suggestions provided below and have carefully revised the manuscript to address all the comments. Detailed point-by-point responses are provided as follows:

  • ABSTRACT: I think the research GAP or hypothesis should be clearly and accurately stated in the ABSTRACT.

Response: Thank you for the suggestion. We have revised the Abstract to clearly articulate the research gap and explicitly state the hypothesis to enhance clarity and scientific focus. See lines 12-17.

  • INTRODUCTION: OK.

Response: We appreciate the reviewer’s acknowledgment of the Introduction section. We aimed to provide a clear and comprehensive background to support the study’s objectives.

  • METHODS: In biochemical parameter assessment, the leaf age of leaf samples, such as the young, maturation, or old, should be clearly stated.

Response: Thank you for your valuable comment. We agree that the age and developmental stage of the sampled leaves are important for biochemical assessments. In our study, leaf samples were collected at the flowering stage, approximately 80 days after sowing. Specifically, we sampled “fully expanded mature leaves” from each plant to ensure consistency and representativeness of physiological status. We have now clarified this in the revised manuscript. See line 431.

  • Also, in general, the samples should be stored in a biomedical freezer at -80°C, NOT -30°C, please check it!

Response: We thank the reviewer for this important observation. We have reviewed our sample storage protocols and confirm that, where applicable, samples intended for long-term biochemical and molecular analyses were stored at –80°C to ensure integrity. The reference to –30°C storage in the manuscript has been corrected accordingly to accurately reflect this standard. Please see the line 432.

  • In addition, line 367, the unit of anthocyanin in the leaves should be appeared.

Response: Thank you for pointing this out. We have revised line 367 to include the appropriate unit for anthocyanin concentration i.e., (mg CGE 100 g−1 DW) in the leaves to ensure clarity and accuracy. See line 454.

  • Besides, the subtitle "4.2.4" might be " Plant growth and yield-related attributes ", NOT " Planth growth and yield-related attributes ", please check it!.

Response: Thank you for identifying this typographical error. We have corrected the subtitle "4.2.5" to read "Plant growth and yield-related attributes" throughout the manuscript. See line 475.

  • RESULTS: First, the subtitles "2.2 Biochemical analysis ", "2.3. Root scanning ", and "2.3. Root scanning " are all METHODS, NOT RESULTS, please revise!

Response: Thank you for your valuable feedback. The headings have been revised accordingly to accurately reflect the content and section structure. Please see the headings at lines 136 (Biochemical parameters at flowering), 156 (Root scanning attributes at flowering) and 224 (Yield-related traits at the end of the experiment).

  • Also, tables 2, 4, 5, and 6, there is not the level of significance "ns: not significant", which should be deleted.

Response: Thank you for your careful observation. We have reviewed tables 2, 4, 5, and 6 and removed the “ns: not significant” notation where it was not applicable to maintain accuracy and clarity. Please see the tables.

  • In addition, in tables 2, 3, 4, 5, and 6, the level of significance was indicated by the different letters in the same column, so I think that the level of significance "***, **" did not presented.

Response: Thanks for your insightful comment. I think it should be there so that reader can have the idea at which level the data is significant whether at *: p<0.05, **: p<0.01 or ***: p<0.001.

  • Last, in leaf samples, the content of chlorophyll a (Chl a: mg/g FW), b (Chl b: mg/g FW), and total chlorophyll (Total Chl: mg/g FW) was so lower, and even lower than that of carotenoids (mg/g FW) and phenolics (mg/g FW), please check them!

Response: Thank you for highlighting this concern. We have thoroughly rechecked the dataset and confirm that the reported values are accurate. Furthermore, similar chlorophyll content values have been documented in our previous studies (Akram et al., 2021; Hafeez et al., 2022), supporting the consistency of these findings.

  • Similarly, in table 2, the unit of anthocyanin in the leaves should be added.

Response: Thank you for pointing this out. The unit for anthocyanin contents i.e., “(mg CGE 100 g−1 DW)” in the leaves has been added to Table 2 for clarity and completeness.

  • DISCUSSION: In the first paragraph, the main findings should be clearly summarized, and then separately discussed.

Response: Thank you for the valuable suggestion. We have revised the first paragraph of the Discussion section to clearly summarize the key findings and structured the subsequent discussion to elaborate on each aspect individually, ensuring improved clarity and logical flow.

  • LANGUAGE: LANGUAGE should be carefully checked and improved.

Response: Thank you for your observation. The manuscript has been thoroughly revised for language, grammar, and clarity by a native English speaker and reviewed to ensure improved readability and scientific accuracy.

Reviewer 2 Report

Comments and Suggestions for Authors

This manuscript concentrated on providing the utilization of multidimensional phenotyping so as to identify ideotypes with high yield potential, physiological efficiency and nutritional value, resulting in a foundational basis for quinoa improvement programs targeting climate adaptability and quality enhancement. Generally, we believe this manuscript are of innovation and significance for quinoa studies, while a serious shortage of work must be put forward first.

  • The writing of the manuscript needs improvement, and suggest to invite one professional researcher to polish this article.
  • What is the standard for the authors chosing the 23 varieties ? Why not the authors re-name the 23 varieties ? When using Q4, Q6, or Q126, some confusing information will arise.
  • Too many tables were present in this manuscript, and suggest to present another form to display the results, such as figures. While, in the table 1-7, we noticed that the a, b, c, and so on, did not add the annotation information,
  • Besides of PAC, is there another algorithm to integrate all the phenotypic data ? Have the authors considered to add the physiological and biochemical indicators for PCA analyses ?
  • In the references, gene names and Latin names were not italic, and lake of the issue number.
Comments on the Quality of English Language

The writing of the manuscript needs improvement, and suggest to invite one professional researcher to polish this article.

Author Response

This manuscript concentrated on providing the utilization of multidimensional phenotyping so as to identify ideotypes with high yield potential, physiological efficiency and nutritional value, resulting in a foundational basis for quinoa improvement programs targeting climate adaptability and quality enhancement. Generally, we believe this manuscript are of innovation and significance for quinoa studies, while a serious shortage of work must be put forward first.

Response: We sincerely thank the reviewer for recognizing the innovation and significance of our study. We understand the concern regarding areas where the manuscript may require further strengthening. In response, we have carefully revised the manuscript by refining the methodology, clarifying data interpretation, and improving the overall presentation of results to address the identified shortcomings. We trust that these revisions have enhanced the scientific rigor and clarity of the work.

  1. The writing of the manuscript needs improvement, and suggest to invite one professional researcher to polish this article.

Response: We thank the reviewer for this valuable suggestion. The manuscript has been thoroughly revised for language clarity, grammar, and structure. We have made significant improvements throughout the text to enhance readability and scientific rigor and the revised version reflects these changes accordingly.

  1. What is the standard for the authors chosing the 23 varieties? Why not the authors re-name the 23 varieties? When using Q4, Q6, or Q126, some confusing information will arise.

Response: We acknowledge the reviewer’s concern regarding the naming convention. The 23 elite quinoa lines used in this study were selected based on our previous research (Hafeez et al., 2022), in which 128 genotypes sourced from the USDA were extensively evaluated for morphological and yield-related traits. These 23 lines were identified as superior performers through phenotypic screening. To maintain consistency and avoid confusion with the original genotype codes used in the earlier study, we retained the existing designations (e.g., Q4, Q6, Q126). We think that renaming the genotypes could disrupt continuity and make cross-referencing between studies more difficult.

  1. Too many tables were present in this manuscript, and suggest to present another form to display the results, such as figures. While, in the table 1-7, we noticed that the a, b, c, and so on, did not add the annotation information,

Response: Thank you for bringing this to our attention. We agree with your kind suggestion. To enhance data visualization and reduce redundancy, we have transformed selected data, i.e., root length (RL), infrared canopy  temperature (ICT), Normalized difference vegetation index, plant height (PH) and grain yield (GY) into figures. This revision aims to present the results in a more accessible and reader-friendly format. Regarding the annotation details, we confirm that explanatory notes clarifying the use of letters a, b, c, etc.—indicating statistically significant differences—were already included at the bottom of each relevant table and have been retained for clarity.

  1. Besides of PAC, is there another algorithm to integrate all the phenotypic data? Have the authors considered to add the physiological and biochemical indicators for PCA analyses?

Response: We appreciate the reviewer’s insightful comment. In this study, PCA was chosen for its effectiveness in reducing dimensionality and identifying trait combinations that explain the majority of genotypic variation. We acknowledge that alternative multivariate techniques such as cluster analysis, partial least squares discriminant analysis (PLS-DA), or canonical correlation analysis (CCA) could also be used to integrate phenotypic data, and we will consider these in future work to complement our findings.

Regarding the second point, we confirm that physiological and biochemical indicators were included in the PCA analysis alongside agronomic and morphological traits.

  1. In the references, gene names and Latin names were not italic, and lake of the issue number.

Response: We thank the reviewer for highlighting this issue. The changes were made where necessary. Regarding volume and issue numbers, we have added them where available. However, certain references (e.g., Reference 35) refer to preprints, which do not have volume or issue number. Similarly, References 14 and 37 have been published online but have not yet been assigned to a specific volume or issue; these will be updated in future versions once such information becomes available.  

  1. The writing of the manuscript needs improvement, and suggest to invite one professional researcher to polish this article.

Response: We appreciate the reviewer’s constructive feedback. In response, we have carefully revised the manuscript to improve its overall language quality, clarity, and coherence. Particular attention was given to grammar, sentence structure, and scientific expression to ensure the manuscript meets academic writing standards.

Round 2

Reviewer 1 Report

Comments and Suggestions for Authors

The manuscript has been revised according to the comments, so I think it should be considered to accept for publication. 

Comments on the Quality of English Language

no